# Improving Replication in Endometrial Omics: Understanding the Influence of the Menstrual Cycle

**DOI:** 10.3390/ijms26020857

**Published:** 2025-01-20

**Authors:** Jessica Chung, Peter Adrian Rogers

**Affiliations:** 1Department of Obstetrics and Gynaecology, University of Melbourne, and Gynaecology Research Centre, Royal Women’s Hospital, Parkville, VIC 3052, Australia; jchung@unimelb.edu.au; 2Melbourne Bioinformatics, University of Melbourne, Parkville, VIC 3052, Australia

**Keywords:** endometrium, gene expression, omics, experimental design

## Abstract

The dynamic nature of human endometrial tissue presents unique challenges in analysis. Despite extensive research into endometrial disorders such as endometriosis and infertility, recent systematic reviews have highlighted concerning issues with the reproducibility of omics studies attempting to identify biomarkers. This review examines factors contributing to poor reproducibility in endometrial omics research. Hormonal fluctuations in the menstrual cycle lead to widespread molecular changes in the endometrium, most notably in gene expression profiles. In this review, we examine the variability in omics data due to the menstrual cycle and highlight the importance of accurate menstrual cycle dating for effective statistical modelling. The current standards of endometrial dating lack precision and we make the case for using molecular-based modelling methods to estimate menstrual cycle time for endometrium tissue samples. Additionally, we discuss statistical considerations such as confounding and interaction effects, as well as the importance of recording the detailed and accurate clinical information of patients. By addressing these methodological challenges, we aim to establish more robust and reproducible research practises, increasing the reliability of endometrial omics research and biomarker discovery.

## 1. Introduction

Endometrial function and fertility can be affected by numerous conditions including endometriosis, adenomyosis, and fibroids. A large component of studying these disorders has attempted to identify biomarkers that are associated with the condition in order to understand the aetiology of the disorder and improve diagnosis and treatment. Unfortunately, systematic reviews of endometrial studies have indicated poor reproducibility with a lack of consensus on genes that are differentially expressed between endometrial disorders [1,2].

The complex nature of human endometrial tissue renders it particularly challenging to study. In contrast to the relatively homeostatic state of most tissues, the endometrium undergoes a cyclical process of rapid growth, breakdown, and shedding in response to hormonal changes [3]. This process also involves functional changes, allowing for the implantation of an embryo during a narrow time frame following ovulation [4]. Such dynamism entails rapid changes at the molecular level with variations in gene expression, protein expression, DNA methylation, and metabolites observed across different phases of the menstrual cycle [5,6,7]. Consequently, accurately identifying the menstrual cycle phase from which endometrial tissue samples are taken is critical for effective statistical modelling due to the large role that the menstrual cycle plays in explaining observed variation.

The aim of this review article is to examine underlying factors contributing to the observed lack of reproducibility in endometrial omics studies, with an emphasis on transcriptomics studies, and provide recommendations to mitigate some of these difficulties. We will also discuss the accuracy of current endometrial dating methods and statistical considerations that are relevant to endometrial analyses. Furthermore, we make the case for using molecular methods to obtain menstrual cycle time estimates for endometrial samples and advocate for their inclusion in statistical models.

## 2. Lack of Reproducibility in Omics-Based Endometrium Research

Since the advent of ‘omics’ methods, including genomics, transcriptomics, proteomics, epigenomics, and metabolomics, the concurrent analysis of many thousands of variables has become possible, often to aid biomarker discovery. Many omics-based studies are ’case–control’ studies with the aim of identifying potential mechanisms of action in conditions such as endometriosis and recurrent implantation failure (e.g., [8,9,10,11]). These types of studies have reported many genes to be differentially expressed, suggesting a plethora of biomarkers and biological mechanisms that may be involved in endometrial dysfunction [12]. Often, hundreds or thousands of candidate genes are reported to exhibit substantial differences between cases and controls. However, meta-analyses have described a lack of consensus, modest effect sizes, or conflicting results [1,2,13,14,15].

This lack of replicability in endometrial gene expression research was recently investigated by Walker et al. [2], who found that studies on identifying differentially expressed candidate genes investigating the same endometrial pathology did not form a clear consensus. For example, when examining four studies comparing the mid-secretory endometrium from endometriosis vs. control patients, a total of 1307 candidate genes were identified, but only six genes were found to overlap between at least two studies. Similarly, when examining seven recurrent implantation failure (RIF) studies, a total of 1651 genes were identified between RIF patients and controls, with 41 genes overlapping between at least two studies and only one gene overlapping between at least three studies. The situation became even more concerning when taking into account overlapping genes that change in the opposite direction. For the four endometriosis studies, Walker et al. [2] found nine candidate genes that were identified as changing in opposing directions between at least two studies and, similarly, thirty-three discordant candidate genes between the seven RIF vs. control studies.

This situation is similar to other fields (e.g., psychology [16], cancer biology [17], economics [18]), with each having replication crises where effect sizes tend to be dramatically diminished in replication studies or fail to replicate entirely. In the past, genotype–phenotype associations studies have also been plagued by false positives and overestimated effects, with many researchers at the time directing attention to the problem [19,20,21]. Ioannidis et al. [22] quantified the replication of these early, low-powered studies, finding that only 13 of 1151 candidate loci (1.2%) survived replication using larger GWASs, with over 1000 samples for validation. The authors attributed the high number of published false positives mainly to selective reporting biases. The situation for association studies has improved over time with increased sample sizes, more stringent statistical thresholds, improved methodology, and greater transparency. Concurrently, reviewers have also become more aware of these critical issues, further enhancing the quality and reliability of studies in this field. The widespread adoption of open science practises, particularly the sharing of data and code, has also enabled the synthesis of findings through meta-analyses and the reanalysis of past datasets with new, improved methodology.

Reasons for the lack of replicability have been extensively discussed in the existing literature. Poor statistical methodology, questionable research practises, and the selective reporting of studies with positive results all play a part in the publication of false positives [23,24]. More specifically to endometrial biology, a lack of concordance between studies may partly be explained by small sample sizes, the heterogeneity of disease and patient classifications, and different methodologies used between studies [2]. We also believe that poor analysis methodologies which fail to account for the unique characteristics of endometrial tissue play an important role. Specifically, these methodologies do not sufficiently consider the substantial degree of gene expression variability as the tissue progresses through the menstrual cycle. The following sections will discuss this issue as well as possible ways to mitigate the effects of this variability.

## 3. Gene Expression Variability in the Menstrual Cycle

The unusual biological properties of endometrial tissue present significant methodological challenges to its study. In response to circulating hormones, the endometrium undergoes cycles of tissue repair, estrogen-driven proliferation and angiogenesis, progesterone-driven differentiation and secretion, and progesterone-withdrawal-driven breakdown and shedding approximately once every 28 days over a woman’s reproductive lifetime [3]. Additionally, this dynamic tissue has the ability to entirely change its function in the presence of an implanted blastocyst in order to support pregnancy. These dynamic cyclical changes are unique to the endometrium, setting it apart from other tissues.

In addition to the physical and histological changes in the endometrium, a large proportion of genes also undergo changes in gene expression throughout the cycle [7,25,26,27] (Figure 1). Notably, some genes undergo particularly rapid changes. In a study using single-cell transcriptomic data, Wang et al. [28] were able to identify abrupt and discontinuous transcriptomic activation in epithelial cells at the beginning of the window of implantation, affecting genes such as *PAEP*, *GPX3*, and *CXCL14*. Furthermore, Teh et al. [27], using bulk RNA sequencing, were able to identify thousands of rapidly changing genes that changed over an approximate 24 h window at multiple time points in the cycle.

When analysing endometrial expression data, a large proportion of variance in gene expression can be explained by the varying time points in the menstrual cycle at which samples are collected. Gene expression studies frequently employ principal component analyses (PCAs) to explore and visualise data, where highly dimensional expression data are projected to lower dimensional space. This results in the first principal component (PC1) capturing the most variance in the data, the second principal component (PC2) capturing the most variance that is orthogonal to PC1, and so on. In the absence of extreme batch effects, menstrual cycle timing typically emerges as the dominant source of variation. The pattern is commonly captured in the first two principal components for studies with samples across the whole cycle or in PC1 only for studies examining a small subset of the cycle (e.g., studies with only secretory-phase samples). Figure 2 illustrates this pattern using data from GSE234352 which contain endometrial samples from across the entire menstrual cycle.

When modelling omics data such as gene expression, factors that are not of direct relevance to a research question can be considered to be sources of unwanted variation and may include patient age, the sequencing batch, and menstrual cycle time, for example. These are typically included as covariates in a model because variation that is unaccounted for, and the extra noise conferred, can reduce the statistical power to detect real effects and potentially introduce spurious signals, including via confounding [29]. Since the cycle stage is a major source of variation, it is critical that this information is factored in when analysing endometrial omics data.

Concerningly, many published studies fail to account for cycle timing. In a systematic search of published endometrial datasets, Devesa-Peiro et al. [13] found that among 35 case–control studies, 11 studies (31%) did not record any menstrual cycle phase information at the time of biopsy and 13 studies (37%) collected all samples in either the proliferative or secretory phase with no further subdivision. Previous approaches to handling cycle effects have included conducting separate sub-analyses for different cycle phases (e.g., proliferative, early secretory, mid-secretory), as demonstrated in Burney et al. [30], or attempting to limit samples to a narrow time frame (e.g., the window of implantation), such as in Lucas et al. [31], without explicitly including time as a factor in the statistical analysis.

This importance of proper cycle stage correction and the concomitant increase in statistical power was further illustrated by Devesa-Peiro et al. [13] wherein the authors examined 12 prior endometrial gene expression studies for which data had been deposited in the NCBI Gene Expression Omnibus (GEO) database and that provided the menstrual cycle stage among their metadata. The authors found that an average of 44% more genes were identified after adjusting for cycle stage effects compared to without adjustment. When performing differential expression analysis on endometrial data, taking menstrual cycle time into consideration is vital to resolve disease-specific effects from menstrual cycle effects, enabling the more accurate identification of candidate biomarkers.

## 4. Histological and Hormone-Based Endometrium Dating Methods Lack Precision

Endometrial tissue dating is the process of assigning a menstrual cycle stage or time point for a given tissue sample. The Noyes criteria [32] for endometrial dating represent the current gold standard and rely on the histological assessment of morphological features.

When dating endometrial tissue histologically, samples are typically classified into cycle phases encapsulating several days (e.g., ‘menstrual’, ‘proliferative’, ‘early-secretory’, etc.) [33]. Despite being classified in the same phase, samples can have markedly different gene expression profiles, especially during the early–mid-secretory phase when the endometrium enters the window of implantation [27,28]. It has been argued previously that histological endometrial dating lacks sufficient accuracy to enable the reliable and precise assignment of a cycle day or narrow time interval [34]. Accuracy and interobserver agreement between pathologists can vary, as demonstrated in a study by Duggan et al. [35], which only found a 68% agreement within one day between experienced pathologists when dating secretory post-ovulatory day samples.

Furthermore, while inaccuracies from histological endometrial dating are not uncommon for healthy tissue, the situation is exacerbated by the fact that collected samples may be abnormal in some way due to an underlying condition. Issues can include scant samples with little functional tissue, the inclusion of pathological tissue such as polyps, and the presence of endometritis [36]. A poorly developed secretory endometrium and asynchronous glands, with tissues containing a mix of features from varying points in the menstrual cycle, can also cause ambiguity in dating. In particular, the presence of asynchronous endometrial glands is speculated to be relatively common in women with recurrent reproductive failure [37], adding further difficulty to dating samples for the purpose of researching endometrial disorders.

Hormone-based sample-dating methods have also been used to detect where a woman is in the cycle, specifically when examining the secretory phase and fertility (e.g., [38,39,40]). In particular, luteinising hormone (LH) typically peaks 10 to 12 h before ovulation with the onset of the surge starting 35 to 44 h prior to ovulation [41]. Detecting LH in urine is a relatively easy task via commercial products such as test strips [42]. However, there is considerable variation in the pattern of LH surges in different women, such as the amplitude of the maximal LH concentration relative to the baseline and the duration of the peak from the onset [43]. Additionally, Park et al. [43] found variability in LH surge configuration, classifying women into three broad categories: spike (a single peak observed, 41.9%), biphasic (two local maxima observed, 44.2%), and plateau (LH levels remained at the peak for 2–3 days, 13.9%). There is also evidence to suggest that an LH surge does not always precede ovulation, with a retrospective cohort study in women who had previously undergone IVF treatment finding that 46.8% had premature LH surges (that did not precede ovulation), and a subset of those (37%) exhibited multiple premature LH surges [44].

Studies may also employ patient-reported cycle time (e.g., counting the number of days since a woman has started her cycle) as part of their methodology to date endometrial tissue. One complication with this method is that women have considerable variability in cycle lengths. Thus, collecting biopsies from women after the same number of days after the onset of menstruation can result in different cycle phases being sampled. For instance, in real-world menstrual cycle data from more than 600,000 menstrual cycles, an estimated 95% of luteal-phase durations ranged between 10.0 and 14.8 days [45]. This consequently affects accuracy when using LH surges to date samples, where studies often collect samples 6–8 days after the LH peak to target the window of implantation. Combined with the inherent difficulties of detecting the LH peak, this method of dating is likely to be relatively inaccurate.

## 5. Molecular Methods for Estimating Menstrual Cycle Time

More recently, endometrial dating approaches based on gene expression have been proposed [26,27,46]. These molecular dating approaches have the advantage of not requiring a trained pathologist and also being more reproducible and less subjective, given that variation between the assessments of different pathologists is common [34,35,47]. A recent development is the endest R package [27] which estimates time in the menstrual cycle in silico using gene expression data from endometrial biopsies. To develop this method, the authors used 266 histologically dated endometrial samples to characterise the gene expression of all genes throughout the menstrual cycle. Through iterative refinement, the categorical classifications were transformed into a continuous numerical scale and the samples were reassigned to the time point that had the best explanatory power. In essence, this method works by comparing the expression of all genes to the expected expression throughout the menstrual cycle and minimises a loss function to return a time between 0 and 100 that best fits the observed data. To use this method, users must supply either the RNA-seq expression or microarray expression values of their endometrium samples.

Another available computational procedure that can generate an estimate for menstrual cycle time is EndoTime [48], where users supply their own data to train an iterative model. Using RTq-PCR, the authors were able to use the expression levels of six genes to build a model for luteal-phase cycle estimation on a continuous numerical scale. As this procedure is generalisable, users have freedom to employ their preferred gene expression technology and select relevant target genes; however, they must build their own model using labelled training data.

With every endometrial sample assigned a numerical time value, a large proportion of observed gene variability can be explained by this single covariate. Teh et al. [27] showed that when cycle time was modelled as a spline across the whole menstrual cycle, 50% of gene expression models had an R-squared value above 0.19 and 30% of models had an R-squared value above 0.36. This molecular time also demonstrated how rapidly genes can change during certain parts of the menstrual cycle. Figure 3 shows examples of genes that change rapidly in the cycle and have a high proportion of variance explained by the menstrual cycle.

## 6. Advantages of Molecular Dating Methods over Histological and Hormone-Based Methods

The availability of different endometrial dating methods raises questions about the relative accuracy of each. However, this is a difficult question to answer as there is no ground truth available. Which method is ‘best’ may also be dependent on the specific question being asked. Histological dating is the long-established gold standard and is independent from omics data that are analysed. Some may argue that endometrial tissue structure and cellular organisation best characterises the menstrual stage of a tissue sample. On the other hand, it could be asserted that the molecular phenotype is the best representation of what characterises a tissue sample as it can more accurately reflect molecular function and biological processes. Differences in molecular phenotyping may not be detectable under a microscope and molecular quantification can provide more fine-grained estimates.

In the context of differential gene expression, molecular staging is typically superior to histology dating. Capturing as much variation as possible is ideal in order to model the data and maximise statistical power. Consider an RNA-seq endometrium study where the disease state is the variable being studied where all samples are collected during the secretory phase. Any variation observed that is not due to the disease state can be seen as unwanted variation which would be ideally accounted for with covariates. If a PCA plot shows concordance between the position and the estimated time, this indicates that a noteworthy amount of variation can be explained by the estimated time (see Box 1 for examples). A PCA plot showing good concordance with histology dating yet stronger concordance with molecular dating indicates that relying solely on histology dating may leave additional unexplained variance in downstream analyses, variance that could be otherwise captured by using molecular dating. This pattern is likely to emerge even in studies where all samples are collected within a narrow time frame due to how rapidly gene expression can change in the menstrual cycle.

With molecular dating estimates being derived from omics data, the ability to have multiple estimates from independent methods for a single endometrial sample becomes more feasible. One advantage of having multiple dating estimates is the ability to verify consistency. When estimates diverge significantly, discordant samples can be flagged to identify underlying abnormalities such as an asynchronous endometrium or potential sample mix-ups.

Box 1.Comparing provided menstrual cycle information to molecular dating in public endometrial datasets.NCBI’s Gene Expression Omnibus (GEO) database is a public repository containing many thousands of endometrium biopsy samples from transcriptomics studies. When depositing data into GEO, authors typically include sample-associated metadata such as age, BMI, disease status, and some measure of cycle stage. With known issues surrounding accuracy, it follows to question how accurate the cycle stage metadata are for these public datasets. While estimating true accuracy is not possible due to the absence of definitive truth, we can compare the study-provided cycle time with an estimate of the cycle time based on gene expression data [27] and observe concordance across different studies and measures of menstrual cycle time. Table 1 lists nine RNA-seq studies from the GEO database that examined eutopic endometrial tissue and Figure 4 illustrates differences between the provided cycle dating and cycle time es-timated by molecular dating for these studies, with the molecular dating showing greater con-cordance in sample positioning in the PCA space.

**Table 1 ijms-26-00857-t001:** Nine RNA-seq studies from the GEO database that examined eutopic endometrial tissue. The majority of studies provided the LH+ day in their sample metadata, while two studies did not provide cycle stage information.

GEO Series	Number of Samples	Cycle Dating Provided	Reference
GSE106602	70	LH+ day (2, 7, 8)	[49]
GSE98386	40	LH+ day (2, 8)	[50]
GSE65099	20	LH+ day (6–10)	[31]
GSE102131	20	LH+ day (6–10)	N/A
GSE185392	20	LH+ day (6–9)	[9]
GSE180485	20	LH+ day (7–9)	[48]
GSE132711	20	Proliferative or mid-secretory	[51]
GSE172381	40	None	[52]
GSE134056	38	None	[53]

**Figure 4 ijms-26-00857-f004:**
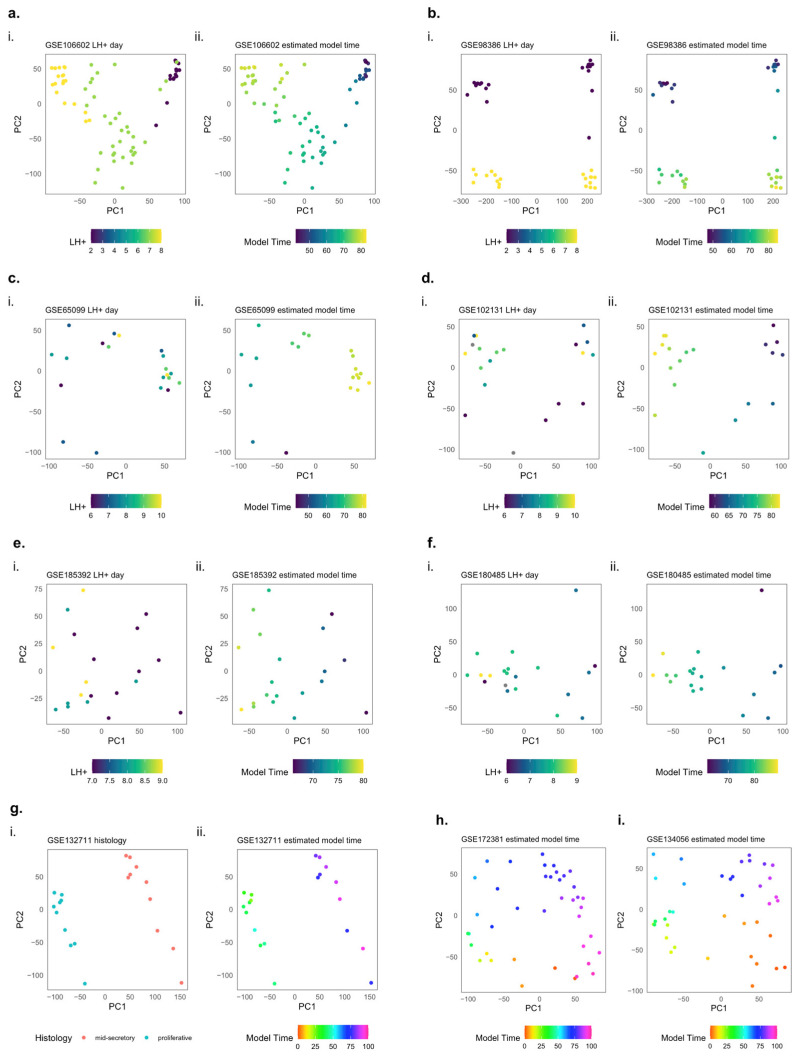
PCA plots from nine RNA-seq studies from the GEO database. Count data were obtained from the GREIN (GEO RNA-seq Experiments Interactive Navigator) platform [54], and molecular model time was obtained using the endest R package [27]. Samples were plotted using the study-provided cycle time (Figure 4**ai**,**bi**,**ci**,**di**,**ei**,**fi**,**gi**) and molecular time (Figure 4**aii**,**bii**,**cii**,**dii**,**eii**,**fii**,**gii**). Two studies had no provided cycle time in their sample metadata and were plotted using molecular time only (Figure 4**h**,**i**). The PCA plots reveal that the variance explained by the first two principal components generally has a stronger concordance with the molecular model timing compared to histological or LH+ dating.

## 7. Other Statistical Considerations for Menstrual Cycle Effects

### 7.1. Sample Size and Chance Confounding

Endometrial samples are typically obtained via invasive biopsy procedures, which can make sample collection difficult, leading to studies with small sample sizes. These reduced sample sizes diminish the statistical power of studies; by definition, limiting the potential to detect differentially expressed genes with high confidence. Many case–control studies attempt to limit sample collection to a specific phase (e.g., [55,56,57]). Chance confounding, where covariates are imbalanced due to chance, is more likely to occur with small sample sizes [58]. Severe imbalances, such as one group containing all earlier samples and the other group containing all later samples, can result in a differential expression analysis between groups mistakenly identifying genes that change due to normal menstrual cycle effects rather than the condition being studied (Figure 5).

Overestimated effect sizes are also more common with small sample sizes due to higher amounts of sampling variability which negatively impacts replicability. Early microarray studies typically had very few patients (e.g., [8,30,57]), and while some studies have been cited hundreds of times, there is still no clear consensus for a set of validated biomarkers that can distinguish between endometrial conditions [2,14]. Small sample sizes, combined with the previously discussed complication of large amounts of gene variation due to cycle stage effects, lead to an increased susceptibility to mistaking noise for signals and publishing flawed results.

### 7.2. Confounding Variables

Many omics studies are observational case–control studies and employ some form of multivariate regression analysis. A critical step in statistical modelling is the careful selection of which biological factors to include as covariates, commonly referred to as ‘controlling for a variable’. These may include factors such as patient age, weight, medications, disease comorbidities, and the time in the menstrual cycle when the biopsy was taken. Ideally, the decision to include or exclude a variable in a statistical analysis involves considering the causal pathways that generate gene expression observations related to the endometrial disorder and therefore requires domain expertise. Care must be taken to not ‘overadjust’ when including covariates in models as this can reduce the effects of factors of interest or introduce bias [60]. Frameworks such as Pearl’s back-door criterion [61] can be employed to evaluate which variables to use as controls, while guidelines containing numerous examples can be found in resources such as those developed by Cinelli et al. [62].

### 7.3. Interaction Effects

In addition to confounding effects, interaction effects between the disorder being studied and other variables should be considered. Interaction effects refer to situations where the effects of two variables are not independent from each other and have varying effects depending on their combination. For instance, an endometrial disorder may have an interaction effect with the menstrual cycle stage, leading to different gene expression changes in different cycle phases. For example, Burney et al. [30] identified the dysregulated gene expression of genes involved in cellular proliferation during the early secretory phase in the endometrium of women with endometriosis which was not seen in other phases of the menstrual cycle. Interaction effects can be addressed by including interaction terms in the statistical model or stratifying the samples by the menstrual cycle stage and disease state and performing contrasts between the relevant groups.

### 7.4. Discrete vs. Continuous Cycle Time Modelling

As discussed previously, menstrual cycle data are often recorded as a cycle phase when estimated histologically, which spans several ‘cycle days’, during which many tissue and gene expression changes can occur. This lack of resolution is suboptimal. Time within the menstrual cycle is continuous, and as such, discretising the variable into blocks results in a loss of information that can have adverse effects on statistical power [63]. For example, a sample in the proliferative stage might be closer to other early-secretory-stage samples than other proliferative stage samples but still be classified as proliferative. Examining a PCA plot of the data and identifying outliers can mitigate some of this effect; however, it can also introduce researcher degrees of freedom (where certain decisions can be made to bias towards more favourable results) in classifying which samples belong in which cycle group. On the other hand, one advantage of recording menstrual cycle time as a continuous numerical value allows it to be used in regression models as a numerical variable, with the option of modelling time functions as non-linear, such as by using polynomial or spline functions (see Box 2 for implementation examples). Since gene changes throughout the menstrual cycle are conspicuously non-linear (Figure 3), these methods have considerable utility in explaining observed variance, especially when modelling longer timescales across different parts of the cycle.

Lastly, the amount of variation that is observed for a particular gene may not be constant throughout the menstrual cycle; there may be points in the cycle where expression varies greatly across a cohort. Such behaviour violates homoskedasticity assumptions for linear methods such as *t*-tests and ANOVAs and should be kept in mind if working with gene expression data across the whole menstrual cycle.

Box 2.Accounting for cycle effects in linear models.Differential expression analysis is commonly performed using the R programming language [64] using packages such as Limma [65], edgeR [66], and DESeq2 [67]. These methods all require a model formula to set up the statistical model. Here, we create a design matrix which is the approach for model specification in Limma and edgeR. In these examples, ‘endo’ is a binary categorical variable that contains the disease state (case or control) for endometriosis and ‘age’ is a numerical variable.If ‘cycle_stage’ is a categorical variable (which includes phases such as proliferative, early secretory, and mid-secretory), we can simply include it in the model formula: design <- model.matrix(~endo + age + cycle_stage)Where ‘cycle_time’ is a numeric variable, we can simply include it in the model if we want to model the variable as a linear effect. Alternatively, we can use the splines R package to model cycle time as a curve. The ns() function can generate a B-spline basis matrix for use in the design matrix, where ‘df’ is the degree of freedom to control the complexity of the curve: design <- model.matrix(~endo + age + ns(cycle_time, df))Polynomial modelling can also be performed using the base poly() function where ‘degree’ determines the degree of the polynomial:design <- model.matrix(~endo + age + poly(cycle_time, degree))Cyclic splines can be used if endometrial samples span the whole menstrual cycle as cyclic splines can account for the biological continuity between the late secretory phase and the beginning of the menstrual phase. The cSplineDes() function from the mgcv R package (≥v1.7) [68] can produce a cyclic B-spline basis that allows for this modelling. Similarly to specifying the degrees of freedom, we supply the location of the spline knots as a vector. We must also remove a column from the basis to remove linear dependence in the model: X <- cSplineDes(cycle_time, knots)
design <- model.matrix(~endo + age + X[,-ncol(X)])
Detailed guidance on spline modelling and parameter specifications can be found in the documentation of the respective R packages.

## 8. Replicability Issues in DGE Analysis

The type of data produced by genomic studies is particularly prone to problems in statistical analysis that exacerbate irreproducibility. Analyses of datasets with high dimensionality, where large numbers of potential response and explanatory variables are measured, are prone to becoming ‘fishing expeditions’. Often, thousands of hypotheses are tested in the hope of finding a significant result, without necessarily considering the number of different statistical tests that have been applied along the way or conducting appropriate multiple-testing adjustment. This can be particularly challenging for endometrial studies in which tissue collection occurs often from multiple different menstrual cycle stages that are dissimilar enough to justify separate sub-analyses rather than one combined analysis (e.g., [30,69]). When performed correctly, statistical methods such as multiple hypothesis testing correction can mitigate reporting incorrect results by reducing false positive rates. However, multiple comparisons are not always as simple as counting the number of statistical tests that are performed and adjusting for this.

Gelman and Loken [70] describe ‘The garden of forking paths’ where many informal, post hoc decisions made in the data collection and analysis process are performed after observing data and can cause inadvertent problems for replicability. For example, a researcher could choose to perform a different statistical test based on initial observations, exclude specific data points, include different variables for a regression, or decide to look for interaction effects when main effects do not appear significant. This leads to unintentional effects that resemble ‘p-hacking’, where many statistical tests are performed and only significant results are reported. One way to mitigate this is to preregister hypotheses and analysis methods before observing the data, which would reduce post hoc decisions. Preregistration has been empirically shown to increase the number of null results, with Kaplan and Irvin [71] observing that the number of positive results from 55 large RCTs decreased from 57% to 8% under the requirement for preregistration and transparency measures.

The current paradigm of Null Hypothesis Significance testing (NHST) under the frequentist framework can be seen as another factor contributing to poor reproducibility and has been often criticised [72,73,74]. NHST is the default method of learning statistics in undergraduate statistical education, the practice of which typically consists of setting up a null hypothesis, calculating a *p*-value with a statistical test, and then rejecting the null hypothesis if the *p*-value is below a certain threshold, often *p* < 0.05. One common charge levelled at NHST is that it asks the wrong question, leading to misinterpretation [74]. Researchers are often interested in the probability of their hypothesis being true given the data, rather than the probability of observing the data (or more extreme) under the null hypothesis. Another important point is that *p*-values also do not convey effect sizes. An emphasis on searching for statistical significance (“stargazing”) can lose sight of the biological consequence attached to statistically significant findings. With very large datasets, NHST allows minuscule effect sizes to become detectable, perhaps with extremely low *p*-values, even if they are biologically inconsequential [74]. Moreover, biological systems are complex with often a large number of interacting variables; thus, the presence of confounding variables is likely. Simulations by Bruns and Ioannidis [75] have demonstrated that uncorrected confounding can lead to false positive results that are indistinguishable from true effects.

Many researchers have advocated using Bayesian methods as an alternative to NHST, owing to Bayesian methods having numerous advantages [73,74,76]. A key feature of the Bayesian framework is the incorporation of existing knowledge into statistical models through the specification of priors, which can lead to more accurate inferences with limited or noisy data. Bayesian methods also result in a probability distribution of parameters, offering a more accurate description of the uncertainty around a measurement. Additionally, hierarchical models can be constructed to model complex scenarios and handle missing data and measurement uncertainty through probabilistic modelling. The growing accessibility of powerful computing resources and the development of probabilistic programming languages such as Stan [77] have facilitated the adoption of these statistical models that use simulation-based methods.

Many of the challenges previously mentioned impact the replicability of endometrial omics analysis, but there also exists a widespread bias in DGE publications as a whole. Päll et al. [78] examined differential analysis from RNA-seq data in the NCBI GEO database, estimating a 59% upper limit of reproducibility (i.e., obtaining identical results when using the original workflow and downloaded data), with more recent studies being more reproducible. Analysis of *p*-value distributions from each experiment revealed that only 23% of experiments had a theoretically expected *p*-value distribution, again with an improving trend over time. The mean proportion of null effects was also shown to increase over time, indicating that genes were less likely to be identified as differentially expressed between conditions in more recent experiments. Analysis software used in respective experiments was heavily associated with both *p*-value distribution and proportion null effects, implying that part of the bias was driven by the chosen methodology.

## 9. Pathologies, Comorbidities, and Accurate Diagnoses

Case–control studies are commonly employed in omics-based studies, grouping all patients with the pathology in one group and all controls in the other. Classifying patients into analysis groups is a critical part of research and is made difficult by the heterogeneity of endometrial disorders. Placing all women with the disorder into a single, homogeneous ‘case’ group may not be optimal for analysis. For example, the symptoms of women with endometriosis are highly variable, with 30–50% of endometriosis patients being infertile, 40–50% experiencing chronic pelvic pain, and 20–25% being asymptomatic [79]. This multifactorial presentation is likely to be reflected in molecular phenotyping, wherein various pathways are differentially activated in the presence of different symptoms, yielding considerable variability. The application of simple case and control groups without accounting for sub-types would be expected to decrease the statistical power to resolve differential expression signals.

Severe cases of endometriosis have been shown to have greater ‘genetic loading’ in GWAS analyses [80,81], as indicated by an increased number of SNPs conferring higher risk. After excluding mild endometriosis samples, Nyholt et al. [80] were able to discover additional SNPs associated with endometriosis, despite a smaller sample size. Infertility-related disorders may also have diverse aetiologies. Recurrent implantation failure, for example, is hypothesised to have two possible endometrial causes: the displacement of the window of implantation and disruption of the window due to pathologies, both of which could coexist in the same patient [82]. These two aetiologies are likely to confer different gene signatures and pathway involvement. As discussed above for endometriosis, the simple grouping of all infertile samples to form a homogeneous collection regardless of presentation would be expected to reduce the likelihood of identifying differentially expressed genes; there are likely to be subtypes (e.g., the displacement of the window of implantation) of subtypes (e.g., infertile).

There exists a trade-off between defining homogeneous disease sub-groups and sample size because the class (sub-group) sample size is reduced with stricter criteria for patient inclusion. While smaller-sized classes generally tend to reduce statistical power, in many cases, more homogenous classes would be expected to more than offset the sample-size effect by reducing variability within groups, thereby allowing biological signals specific to a particular severity or subtype of disease to be revealed; essentially, class-specific signals would become more ‘focused’ [83].

Comorbidities in endometrial disorders are common and represent another important consideration when performing experimental design and analyses. The presence of endometriosis has been associated with an increased risk of other gynecological conditions such as adenomyosis [84], uterine fibroids [85] and ovarian cancers [86]. Non-gynecological comorbidities such as interstitial cystitis and irritable bowel syndrome further complicate matters as they have overlapping pain symptoms caused by shared mechanistic features [87]. These comorbidities can confer statistically confounding effects and, as such, it is vital that they are recorded.

Many endometrial pathologies share overlapping symptoms, such as pelvic pain, abnormal uterine bleeding, and infertility [88], which, combined with high symptom heterogeneity, can contribute to difficulties in diagnosis. A retrospective study by Orlov and Jokubikiene [89] found that half of women displaying endometriosis-associated symptoms had no abnormal transvaginal ultrasound findings. The definitive diagnosis of endometriosis requires surgical and histological visualisation and, on average, diagnosis takes approximately 7 years from the onset of symptoms [90,91]. Similarly, the average time to diagnosis for symptomatic uterine fibroids is approximately 3.6 years [92].

Comorbidities can confound diagnosis, making causal attribution to symptoms difficult. Even in the case of a single observed pathology, a diagnostician cannot ascribe a symptom to that pathology with high confidence. Diagnoses can be missed and pathologies may coexist with symptoms without being responsible for them. To further illustrate the complexity around the diagnosis of endometrial conditions, it has been observed that in over a third of cases of chronic pelvic pain symptoms reappear or persist after diagnosis and treatment [93]. This is also reflected in the non-response rate of endometriosis to laparoscopic surgery, which ranges from 20% to 38% [94,95].

All bioinformatic analyses are downstream from biological classification. If diagnosis and classification are prone to error, results will be noisy and confer diminished statistical power. Perhaps a better approach to analysis might be to focus on symptoms instead of pathologies. The reason for this is twofold. Firstly, diagnosis is difficult, and symptom classification may lead to cleaner, more homogenous groups. Secondly, many women with pathologies have normal endometrial function, despite the presence of pathologies, and the symptoms, such as pain and infertility, are of most concern to patients. Practically, disorders such as endometriosis and fibroids do not require treatment in asymptomatic cases; thus, focusing on symptoms may better target clinical interventions and efforts towards discovering future treatments.

## 10. Conclusions

The current state of endometrial omics research suffers from a lack of replicability. A major reason for this is a failure of properly accounting for the menstrual cycle being a major source of variation. It is critical that a measure of the menstrual cycle stage is included in statistical models when modelling endometrial omics data to mitigate possible confounding and increase statistical power to find real effects. The current standards of endometrial omics methodology are inadequate and there is no justification for omitting menstrual cycle time from future analyses.

Histological and hormone-based methods are commonly used for endometrial dating; however, we recommend using molecular methods (such as the method implemented in the endest R package [27]) for estimating cycle time in which gene expression data from endometrial tissue are used to obtain an estimate. Molecular methods typically explain a higher proportion of observed variance and have the added advantage of generating continuous variable values, allowing cycle time to be modelled as non-linear in regression models (e.g., using splines or polynomial functions).

To improve the replicability of research findings, good statistical analysis is not enough; good data collection is also needed. While clean case–control groups are ideal, such groups are not always feasible. Consequently, it is important to record any phenotypic observation that may impact gene expression such as the presence of endometritis and other abnormalities. To further enhance replicability, pre-registration is encouraged. Decisions such as cycle stage sub-analyses and defining breakpoints when dichotomising variables ought to be decided beforehand. Finally, conducting validation studies with new samples is essential to reduce the propagation of inaccurate research findings.

While endometrial tissues present unique challenges with regard to experimental design and bioinformatic analysis, steps can be taken to minimise publishing erroneous conclusions. By considering these factors, researchers can enhance the reproducibility and validity of their findings.

## Figures and Tables

**Figure 1 ijms-26-00857-f001:**
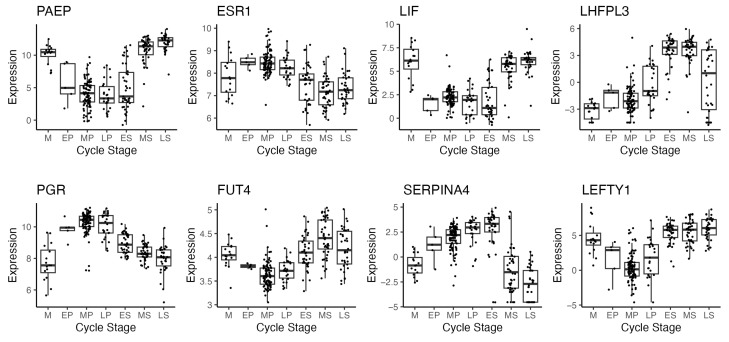
Examples of genes that significantly change in the menstrual cycle. Gene expression from RNA-seq with samples histologically dated from the menstrual (M), early proliferative (EP), mid-proliferative (MP), late proliferative (LP), early secretory (ES), mid-secretory (MS), or late secretory (LS) phase. Each data point represents an endometrial sample taken from a unique patient. Data are from GEO Series GSE234354 [27].

**Figure 2 ijms-26-00857-f002:**
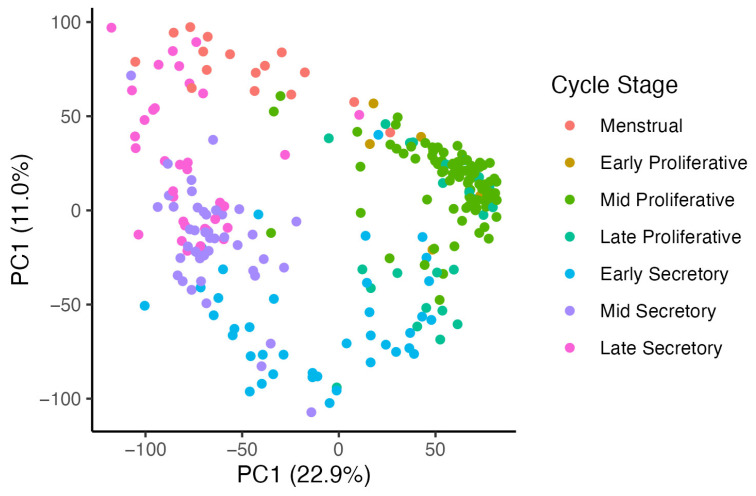
A PCA plot of RNA-seq gene expression data from endometrial tissue samples from across the menstrual cycle. The variance observed in the first two principal components primarily reflects menstrual cycle-related changes in gene expression. Data are from GEO Series GSE234354 [27].

**Figure 3 ijms-26-00857-f003:**
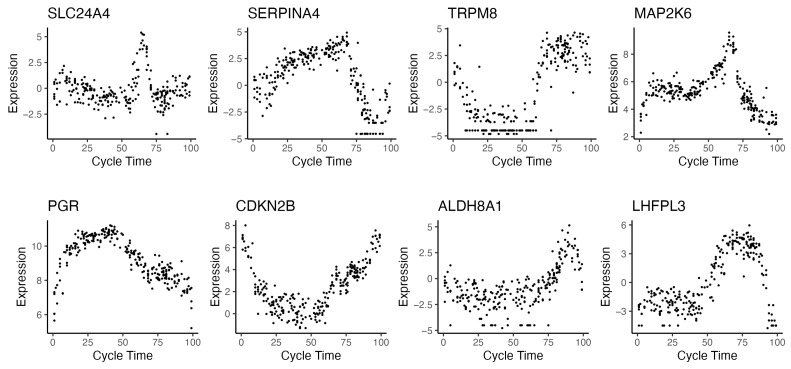
Examples of genes that significantly change expression in the menstrual cycle identified using molecular methods for estimating menstrual cycle time. Each data point represents an endometrial sample taken from a unique patient. Time 0 corresponds to the beginning of menstruation and time 100 corresponds to the conclusion of the secretory phase. Gene expression from RNA-seq with cycle time was estimated using the endest R package [27]. Data are from GEO Series GSE234354 [27].

**Figure 5 ijms-26-00857-f005:**
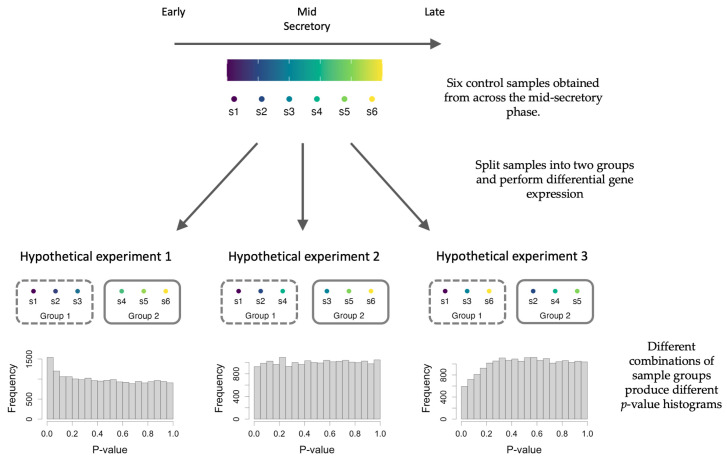
An illustration of the effects of cycle time being confounded with experimental groups. All samples are histologically dated as mid-secretory and are all control samples. The samples are separated into two groups to perform differential gene expression and three different combinations of comparisons are shown. *p*-value histograms are useful for diagnosing issues [59]. Under the null, we expect a flat distribution, as seen in hypothetical experiment 2. The composition of different samples for each experimental group can determine the distribution of *p*-values, e.g., conservative, flat, and anti-conservative distributions. Severe imbalances such as in hypothetical experiment 1 can result in an anti-conservative *p*-value distribution which can result in identifying false positive effects that are due to cycle effects. In a real experiment, it would not be possible to distinguish if the effect is due to the confounding variable or the effect of interest.

## Data Availability

No new data were created or analysed in this study. Data sharing is not applicable to this article.

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
