# Peer review of "Improving Replication in Endometrial Omics: Understanding the Influence of the Menstrual Cycle"

_ijms, 2025, doi:10.3390/ijms26020857_

Round 1

Reviewer 1 Report

Comments and Suggestions for Authors

The manuscript presents a review of factors influencing the reproducibility of endometrial omics research, particularly with respect to gene expression variability across different phases of the menstrual cycle. It highlights methodological challenges and proposes strategies for improving the reliability of research outcomes, including the use of molecular-based modelling to date menstrual cycle stages in endometrial tissue samples. The authors focus on improving experimental design and statistical modelling, aiming to address confounding effects and enhance the precision of future omics studies.

The topic of this manuscript is timely and highly relevant for the field of endometrial research and omics studies. The importance of menstrual cycle variability in gene expression is well-acknowledged, and the review offers valuable insights into improving the replicability of studies in this area. However, there are several areas that require major revisions to ensure clarity, scientific rigor, and comprehensiveness.

Major revisions:

1) Clarification of Methodology Recommendations:

While the authors propose the use of molecular-based methods for menstrual cycle dating, it would be helpful to expand on the specific molecular approaches or techniques that have shown promise, such as gene expression profiling or transcriptomic methods. Detailing the advantages of these methods over traditional hormone-based or histological approaches would strengthen the argument for their implementation in endometrial research.

2) Statistical Considerations:

The manuscript mentions the importance of correcting for menstrual cycle effects in statistical models but does not provide sufficient details on how to implement such corrections. The authors should include specific recommendations for statistical models that could be used to account for cycle-related variability, such as mixed-effects models or spline-based approaches. Additionally, they should discuss how to handle potential confounding variables (e.g., patient age, comorbidities) in a more detailed manner.

3) Data Interpretation and Generalizability:

Although the review emphasizes the variability in gene expression across the menstrual cycle, it would benefit from a deeper exploration of how this variability impacts the reproducibility of omics findings. The authors should provide more concrete examples or case studies where cycle variability has led to reproducibility issues. Furthermore, the review could discuss how these issues are addressed in other omics fields (e.g., cancer biology) and whether similar strategies could be adapted for endometrial studies.

4) Impact on Biomarker Discovery:

Minor revisions:

1) Figures and Data Presentation:

The manuscript includes important figures, but some of the figures could benefit from more detailed captions explaining the data sources, methods of analysis, and implications of the results. For example, the PCA plot (Figure 2) would be clearer with a brief description of how the cycle time effects were adjusted and why this adjustment improves the analysis.

2) Literature Comparison:

The manuscript provides a solid foundation for improving omics research in endometrial tissue, but it could benefit from a more thorough comparison to existing literature. For example, the authors should discuss the current best practices for menstrual cycle correction in other omics fields and how they might be applied or modified for endometrial research.

3) Ethical Considerations and Data Sharing:

Given the importance of accurate clinical information and patient data, the authors should include a more detailed discussion on ethical considerations for omics studies involving human tissue. The manuscript could also mention the importance of data sharing and transparency in omics research to enhance reproducibility.

4) Typographical and Grammatical Errors:

There are a few minor grammatical errors and awkward phrases that should be revised for clarity. For instance, in some sections, sentence structures could be simplified to improve readability.

This manuscript presents an important topic in the field of endometrial omics research. The review provides useful suggestions for improving the reproducibility of studies, especially regarding the critical influence of the menstrual cycle. However, the manuscript requires substantial revisions to clarify the methodology, enhance the statistical discussion, and provide more concrete examples of the challenges in replicating findings. With the proposed revisions, this manuscript has the potential to make a significant contribution to the field.

Recommendations: Major revisions required for methodology details, statistical considerations, and example case studies.

Comments on the Quality of English Language

The manuscript generally has good clarity, but requires minor grammatical and typographical corrections to improve readability and professionalism.

Reviewer 2 Report

Comments and Suggestions for Authors

I would encourage the authors to emphasize the foundational principles of endocrinology, particularly the impossibility of identifying specific biochemical markers in the endometrial cycle that remain unchanged over a five-month period. The metabolic clearance rate (MCR) and production rate(PR) of steroid hormones, which drive endometrial omics variations throughout the menstrual cycle, are influenced by factors such as body temperature and physical activity. These variables introduce significant unpredictability, making it impractical to perform timing analyses before the cascade of biochemical events and subsequent morphological changes take place.
